# "Perugia Upside-Down": A Multimedia Exhibition in Umbria (Central Italy) for Improving Geoheritage and Geotourism in Urban Areas

**Laura Melelli**

Department of Physics and Geology, University of Perugia, 06123 Perugia, Italy; laura.melelli@unipg.it; Tel.: +39-07-5584-9579

**Abstract:** Multimedia materials represent a promising approach to the promotion of geoheritage. Despite geology being normally associated with natural environments, new tendencies are noted towards better knowledge of the "geological reason" for the selection of a location and the development of urban settlements. The urban environment is, in fact, a perfect laboratory for opening the scientific topics to a broad audience. In this paper, the experience of a geological exhibition organized in the city of Perugia (Umbria, central Italy) is discussed, highlighting the SECRET (SEe and CREaTe) for creating an effective dissemination activity. Panels, interactive tools, laboratories, and trekking tours outside the museum are the main activities, which hosted more than eight thousand visitors in a few months. Moreover, the exhibition was the starting point for ongoing projects on geotourism in the city, with important consequences in terms of visibility and financial return.

**Keywords:** geotourism; geoheritage; urban geology

## 1. Introduction

The common idea of geology as a scientific discipline restricted to the natural environment is quite widespread and consolidated. However, increasing attention to the geological investigation of urban areas is growing in the scientific community [1–3]. The establishment of a city always has a geological reason. The situation and the site are the initial starting points. The situation or position is the geographical location related to the surrounding areas, being fundamental for communications, economic relations, and cultural exchanges with other communities. In other words, the position refers to how a place is related to other cities or productive places [4]. The site conditions set the direct relations within the environmental context [4]. The topographic conditions (slope angle values in relation to the possibility of defending against external attacks) as well as the proximity of rivers or the sea and the availability of underground water are the most important criteria for site selection. Moreover, the bedrock composition should support the building material and the possibility to create hypogean cavities for a large number of uses (drainage or water supply, food storage, underground passages, shelters in case of war). The geomorphological conditions, in particular, the evolution of a site in relation to landslides or flooding events, establishes the possibility for the urban fabric to extend in the surrounding areas. A large part of scientific literature is focused on natural hazards in cities [5]. Floods or droughts [6] and their increasing effects due to climate change [7] are one of the topics in this area. Other specific and more local natural hazards, such as volcanic or seismic events, also affect urban areas [8–10].

Presently, an opposite trend is growing in the scientific and administrative environments: The geotouristic approach, where the geological context is a new and promising resource for the touristic and didactic issues in urban areas. Geotourism is the branch of tourism focused on activities,

products, and services related to Earth sciences [11,12] where the subject is the geological component of the natural environment and social context with a high scientific, educational and cultural value. The prefix "geo-" includes "geology, geomorphology", and the natural resources of the landscape, landforms, fossil beds, rocks and minerals, with an emphasis on appreciating the processes that are creating and created such features [11]. Geotourism links the geology as a scientific discipline using objective criteria and scientific methods to tourism, which needs subjective criteria and aesthetic components [13]. Geotourism is the most efficient approach for exporting the scientific contents of the Earth sciences to a wider audience, characterized by a wide spectrum of ages and cultural backgrounds. Geoheritage is the cornerstone of geotourism, that is a category of heritage where the geological component is relevant. Where some areas show characteristics of uniqueness in both the scientific and cultural aspects, they are selected and classified as geosites [14] and geomorphosites [15–17]. A geosite is the best expression of geoheritage but geosites are not always present in some areas and, moreover, their definition is not simply objective. Therefore, in order to export the knowledge derived from Earth sciences to a wider public, it is essential to find the geological component of a landscape also in common features and daily experiences.

Aside from the definitions surrounding the subjects of geotourism, the real challenge is how to communicate this heritage and most of all, how to make it a recreational activity. A huge amount of scientific papers and research activities are devoted to these methods and represent the main vehicle of dissemination for cultural geoheritage [18,19]. This tendency has had exponential growth since 2001 [20]. However, several problems arise for dissemination including the technical scientific language, the geological time scale (millions of years) and the spatial scale varying from extensions of thousands of kilometers to the microscopic scale [13]. Another problem is the high heterogeneity in the tourists involved in geotourism. Differences in age, cultural level, and physical capability may be a serious obstacle for successful dissemination [13]. Finally, the area of interest of geotourism does not equally cover all the branches of knowledge related to Earth sciences. Geomorphology, volcanology, and paleontology are the most exploited in dissemination activities [13] since these subjects investigate more than others the macroscopic effects of geodynamics and are linked to the most fascinating aspects of the geology, recalling spectacular and impressive natural events.

Introducing the idea of the geological component in a city as a strong point of tourist activity is not easy. The traditional approach in visiting and getting to know a city, both for tourists and educational purposes, is generally starting from a historical framework. The geographical introduction, if it is present, it is reduced to a brief paragraph. Moreover, the link between the geographical setting and the human presence is absent in most cases. Improving the geological heritage should be the basis for introducing people to a city. The morphological and hydrographic arrangement is a direct consequence of the geological evolution of the area. The time span considered is much broader, but it is essential information for understanding where, why, and how the local populations made their choices in order to exploit resources and oppose the limits of the territory. To date, the geological parameter in cities is perceived as a risk. Where the geological heritage in situ is not present or well evident, as in some urban areas, a good compromise is represented by the ex situ items, such as museum collections. In dissemination activities, the museum with permanent and temporary exhibitions are one of the most successful possibilities [21]. Nevertheless, except for dinosaurs, volcanoes, and earthquakes, geological matter is not very interesting for non-specialists [21].

In order to stress the idea of the exhibition as a good tool for dissemination activities related to Earth sciences, a geological exhibition was organized in Perugia in 2017 (Umbria, central Italy, Figure 1A).

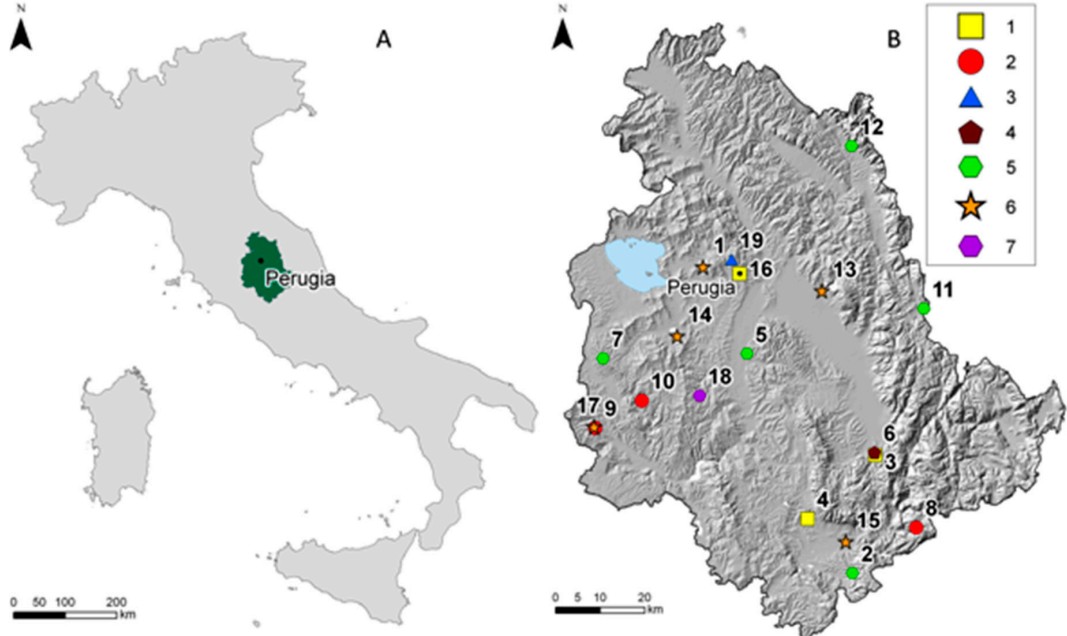

**Figure 1.** (**A**) Location map: the Umbria region in Italy with the Perugia city. (**B**) The Umbria region with the scientific museum already present on the regional territory and dedicated to natural sciences and Earth sciences. The different symbols represent the specialization: (1) Didactics, (2) Geology, (3) Hydrogeology, (4) Mine, (5) Natural Science, (6) Paleontology, (7) Volcanology. The numbers inside the figure refer to Table 1: (1) Antiquarium Museum (Corciano, PG), (2) Civic Museum of Natural History (Stroncone, Terni), (3) Earth Science Laboratory of Spoleto (Spoleto, PG), (4) GeoLab (San Gemini, TR), (5) GSN Gallery of Natural History (Casalina, PG), (6) Morgnano Mines Museum (Spoleto, PG), (7) Museum of Natural History and of Territory (Città della Pieve, PG), (8) Museum of the Apennines (Polino, TR), (9) Museum of the Geological Cicles (Allerona, TR), (10) Museum of the Territory (Parrano, TR), (11) Naturalistic Museum of Colfiorito Park (Colfiorito, PG), (12) Naturalistic Museum of Cucco Mt. and Earth Science (Costacciaro, PG), (13) Paleontological Museum (Assisi, PG), (14) Paleontological Museum (Pietrafitta, PG), (15) Paleontological Museum (Terni, TR), (16) TerraLab (Perugia, PG), (17) The Botanic Palaeontology Centre of the Fossil Forest in Dunarobba, (Allerona, TR), (18) Volcanological Park of San Venanzo (San Venanzo, TR), Water Museum (Perugia, PG). The abbreviation PG is for Perugia, the abbreviation TR is for Terni, Perugia and Terni are the two provinces of the Umbria region.

Umbria is a region with strong evidence of a connection between topography, morphology, and geology, and so is an excellent test area for such dissemination activities. Nineteen museums, with permanent exhibitions focused on some aspects of Earth sciences are already present in the regional territory (Figure 1B). Five of them are focused on paleontological heritage and as many on a wider naturalistic aspect where the geological component is only a part of the exhibition. Three museums are devoted to general aspects of local geology, while one is dedicated to mining activity, one to volcanology and another one to hydrogeology. All these museums offer occasional didactic laboratories but only the remaining three museums have permanent laboratories and exhibitions for didactic purposes. In Table 1, the museums are listed with their specific vocations.

Although Umbria is a small region, it can, therefore, count on a good number of initiatives aimed at divulging geological data. However urban geology has never been the subject of dissemination activities in the region. This paper illustrates the first attempt to do that in Perugia, one of the most important hotspots for cultural initiatives in central Italy and offering a large number of aspects useful for research on urban geology. This paper illustrates in detail the scientific background, the dissemination techniques and the results of this experience.

**Table 1.** Museums present in Umbria. The numbers refer to Figure 1B. Type: D) Didactics, G) Geology, H) Hydrogeology, M) Mine, NS) Natural Science, P) Paleontology, V) Volcanology. The abbreviation PG is for Perugia, the abbreviation TR is for Terni, Perugia and Terni are the two provinces of the Umbria region.

| N. | Name | Location | Type |
|----|------|----------|------|
| 1 | Antiquarium Museum | Corciano (PG) | P |
| 2 | Civic Museum of Natural History | Stroncone (TR) | NS |
| 3 | Earth Science Laboratory of Spoleto | Spoleto (PG) | D |
| 4 | GeoLab | San Gemini (TR) | D |
| 5 | GSN Gallery of Natural History | Casalina (PG) | NS |
| 6 | Morgnano Mines Museum | Spoleto (PG) | M |
| 7 | Museum of Natural History and of Territory | Città della Pieve (PG) | NS |
| 8 | Museum of the Apennines | Polino (TR) | G |
| 9 | Museum of the Geological Cicles | Allerona (TR) | G |
| 10 | Museum of the Territory | Parrano (TR) | G |
| 11 | Naturalistic Museum of Colfiorito Park | Colfiorito (PG) | NS |
| 12 | Naturalistic Museum of Cucco Mt. and Earth Science Laboratory | Costacciaro (PG) | NS |
| 13 | Paleontological Museum | Assisi (PG) | P |
| 14 | Paleontological Museum | Pietrafitta (PG) | P |
| 15 | Paleontological Museum | Terni | P |
| 16 | TerraLab | Perugia | D |
| 17 | The Botanic Palaeontology Centre of the Fossil Forest in Dunarobba | Allerona (TR) | P |
| 18 | Volcanological Park of San Venanzo | San Venanzo (TR) | V |
| 19 | Water Museum | Perugia | H |

## 2. The "Perugia Upside-Down" Exhibition: An Example of Best Practice

Perugia is the capital city of the Umbria region, (central Italy) and is located on a triangular-shaped hill with an areal extent of about 27 km$^2$. The maximum altitude value is about 493 m a.s.l. with a minimum of ca. 200 m along the Tiber River valley, at the bottom of the hill (Figure 2). The hill is distributed along five main ridges spreading from the highest altitude toward NE, E, SSE, SW, and W, separated by several small rivers. The hill of Perugia is made of sediments derived from fluvial and/or lacustrine environments, widespread in the area during the Pliocene and Pleistocene.

In these periods, an extensional tectonic phase, still acting, affected the area and the morphological result of this phase are several intermountain basins bordered by normal faults [22,23]. Perugia is located along the western edge of the Tiberino Basin, the largest basin in Umbria (about 1800 km$^2$) and one of the largest in Central Italy (Figure 2B). The bedrock of the hill of Perugia is made of clastic sediments of different sizes, from blocks and gravels to sands and clays, transported by the rivers flowing from the surrounding mountains and then deposited on the bottom of the intermountain basins. In addition, a drainage network of rivers, swamps, and lakes was widespread along the plain areas inside the basins, covering with new sediments and reshaping the previous deposits. The sedimentary sequence, dated in Perugia from Early to Middle Pleistocene, has variable thicknesses from few to hundreds of meters and is defined as "Perugia Unit" (Figure 2B). The unit is divided into some litofacies according to sedimentary and paleoenvironmental principles. In each of these litofacies some deposits prevail. In the Volumni Litofacies, present in the downtown of the city (lower Pleistocene),

conglomerates and sand are prevalent. The extensional tectonic stress is still acting with the result that the morphological evolution is very dynamic [24,25]. Along the borders of the intermountain basins, the sedimentary sequences are faulted and eroded, resulting in gentle hilly areas. The topographic arrangement, with a higher altitude, compared to the lowlands of the alluvial plain, often covered by stagnant water, guaranteed a healthier environment. In the same time, the gentle slope values along the flanks of the hills allowed an easier connection with roads and cities in comparison to the steep mountain areas [26]. The most important historical cities in Umbria are located on the top of these sedimentary hills and their position and sites are a clear consequence of the geological history of the area. Perugia is a perfect example of this condition and a good test site for urban geology and for the scientific communication of this topic.

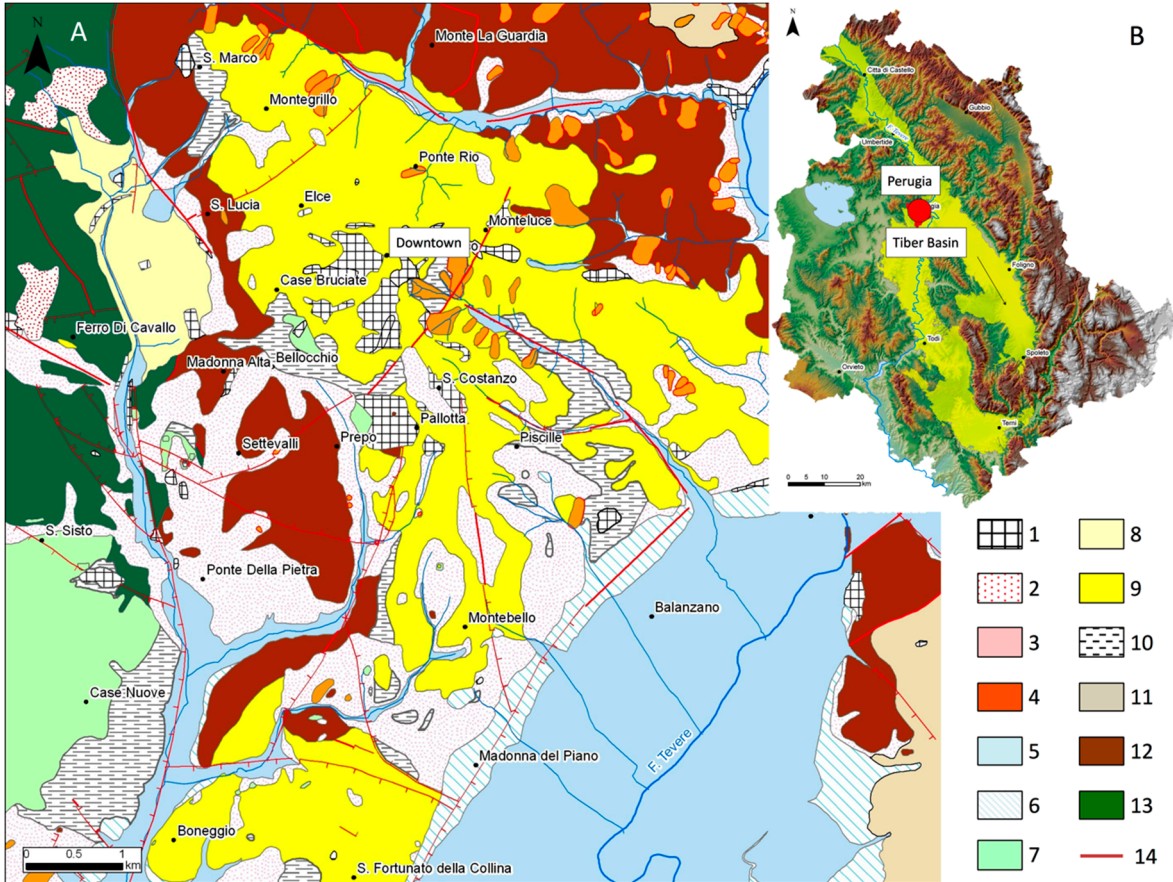

**Figure 2.** (**A**) The geological map of the Perugia city area. (1) Anthropic deposits, (2) Debris (Holocene), (3) Colluvial deposits (Holocene), (4) Landslides, (5) Alluvial deposits (Holocene), (6) Alluvial terrace (Holocene), (7) Perugia Unit, Ellera Litofacies (upper–medium Pleistocene), (8) Perugia Unit, Pian di Massiano Litofacies (medium Pleistocene), (9) Perugia Unit, Volumni Litofacies (lower Pleistocene), (10) Perugia Unit, Ferrini Litofacies (lower Pleistocene), (11) Solfagnano Unit (lower Pleistocene), (12) Terrigenous Complex (Burdigalian–Tortonian), (13) Limestone Complex (upper Trias–lower Miocene). (**B**) The Perugia city in Umbria region and inside the limits of the Tiberino Basin (in yellow).

"Perugia Upside-Down: When the Geology Describes the City" is the title of an exhibition developed by the Department of Physics and Geology of the University of Perugia, inaugurated on 10 November 2017. The exhibit location is the POST Museum (Perugia Officina della Scienza e della Tecnica—Perugia Science and Technology Laboratory, http://www.perugiapost.it), the most important and visited a scientific museum in the city. The exhibition lasted until the spring of 2018 (Figure 3).

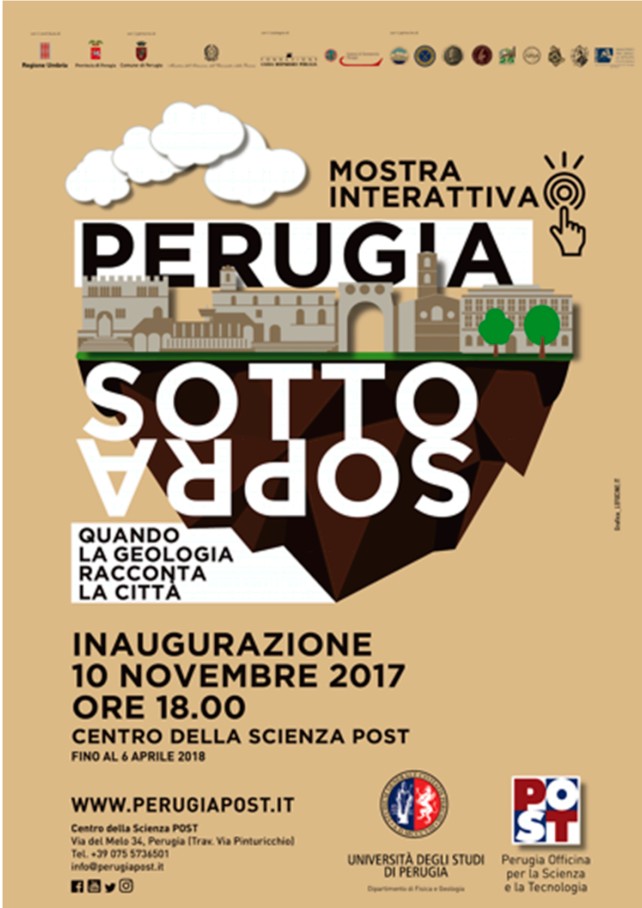

**Figure 3.** The poster used to promote the exhibition. The title was "Perugia Upside-Down: When the Geology Tells the City" (the reproduction of the image is allowed by POST).

*2.1. Methodology*

Exhibition is a common and useful practice in order to attract people [27,28], but the results in terms of the number of visitors or the level of satisfaction are not always encouraging [13]. Problems are related to the traditional method of exposition (samples and description). Taking inspiration from previous experiences, to which others have been added over time as highlighted in the references list [29–34] some already tested items were proposed in the exhibition. One of the most successful approaches is to highlight and illustrate some of the stones used for historical buildings [35,36]. The identification of petrographic characteristics is the starting point to expand the information linked to the paleoenvironmental conditions. Thus, the building stones are snapshots of the geological history of the surrounding areas and, because of their position, visible on the most important buildings of our cities, are instruments always openly available. In addition, some samples show palaeontological features, thus not only the sedimentological or mineralogical data are present but also other added values, introducing a wide range of geological aspects [37,38]. In order to understand the geological composition of lithotypes outcropping under the cities or in the surrounding areas, the building stones are used as a point of interest for urban trekking and they are one of the best expressions of the geoheritage present in urban areas [12]. Historical buildings, bridges or industrial constructions are viewpoint geosites. The considerable height of towers, belfries, industrial sheds (especially if associated with a surrounding topographic arrangement with lower altitude values) offers a unique opportunity to admire the landscape and understand the morphology and spatial location of a city [39,40]. In addition, other aspects are properly used for touristic and didactic purposes. The geomorphological evolution perhaps represents one of the most intriguing cases because, traveling through time, rewinds the morphological evolution and reveals the past, the present

and the future landscape [41,42]. The paleontological heritage, if it was found in the urban area, could be an excellent topic for an exhibition [43,44]. Moreover, when characteristics concerning geology are combined with other fields, such as archaeology, the sites where these characteristics are present at the same time can be excellent targets for geotourism and thematic exhibitions [26,45].

Once the geological aspects are known, the next step is to identify the best solution to disseminate the content. To translate the urban geology from a scientific perspective to well-understood information, some criteria must be satisfied [13]. First of all, is the time interval. Geology is a science that takes into account timescales of up to hundreds of millions of years (the Earth system has been evolving since approximately from 4.5 billion years ago) while the human experience covers at most a few millennia. For common people, thinking in terms of ancient times generally means to enlarge the time perspective up to a few hundred years. The morphological evolution is perceived as something related to an unchangeable system where only the catastrophic events (earthquakes, volcanic eruptions, tsunami, landslides) suggest that the Earth is a dynamic planet. The dynamic equilibrium controlling the surface processes modeling the Earth surface is invisible to the human eyes. This is one of the most relevant difficulties when, for example, the perceived risk is lower than the real risk during natural hazard events. Therefore, in order to communicate the geological evolution of an area, it is fundamental to underline the time spans in relation to human life. The second problem is the four-dimensional perspectives, necessary to a geologist to understand features and events. A geologist often needs to consider a landscape in 3D. In addition, a fourth dimension is needed, considering the structure under the topographic surface too. This means having the skills to consider the landscape from a geographical perspective on the surface of the Earth and keeping the visualization vertical, imagining the removal of the topographic surface as if it was only a thin layer. This skill is not common for people with different knowledge, and therefore one of the greatest efforts that must be made to make scientific communication effective is to introduce a tourist or a student into a "bird's eye" view and then take them below the Earth's surface in the fourth dimension. The third problem is the scale. Geology includes patterns and processes that range from the infinitely large to the infinitely small. To understand Earth dynamics, the observations embrace a spatial framework going from the solar system and beyond until the microscopic observation of the structure of minerals. The challenge is to make clear that these scales are the opposite sides of the same coin and to join the information deriving from different approaches in a unique way. The fourth drawback is the language. Every experience related to scientific communication should translate the scientific language in a common way, using few but unavoidable rules: Concise and without technical terms but exhaustive, in other words, simple but not simplistic. To find the best compromise between complete information avoidance to being incomprehensible and boring is not so obvious. Many experiences attempt to avoid the problem using a glossary, but this is a false solution. It is quite rare that in dissemination activity people are so intensely involved as to seek out clarification each time it is necessary, consulting a glossary. The first reaction is to read a text without fully understanding it. The fifth point, which is specific for urban geology, is to never forget that in the cities the geological evolution is strictly related to human settlement, so never separate the naturalistic aspect from the anthropic one. People may be interested in the natural environment, but they become even more interested if this environment is something that has an impact on their everyday life.

To try to get the best result, the exhibition was structured with a basis of panels in the museum but with several parallel activities with the aim of encouraging visitors to become active subjects, both in the museum and outside: tools, laboratories, trekking tours. To overcome the problems related to the disclosure of a scientific subject listed above, this exhibition was prepared in a synergy between researchers, museum workers, and designers. The researchers have devised the thread of the information flow and prepared the text, the figures, and the theoretical basis for the tools. Moreover, they prepared the laboratories and led the trekking around the cities. The staff museum built the infrastructures to house the material. Most of all, they provided an irreplaceable contribution in simplifying the scientific character of the texts and figures. The designers created the graphics and organized texts and figures on the panels.

Although panels may be the most boring aspect of an exhibition, the cooperation with designers and staff museum guaranteed an amazing and effective final product. The panels were designed following some criteria (Figure 4). The upper section was devoted to the title and to a progressive number showing the path to be followed. At the bottom of the panels, only graphics were present to not force the visitor to bend down. In the middle part, the text was separated in columns with a logical idea, which imposed the public to read the contents going from the left to the right. On the left, only the fundamental concepts were summarized, then moving toward the right side of the panel, other peculiar information was added. The aim was to introduce the visitor to the topic described on the panel, presenting information step by step and giving them the possibility to decide when to finish reading, without losing important information. Figures and photos were always present. Some supplementary boxes were included for explaining technical words or particular geological concepts.

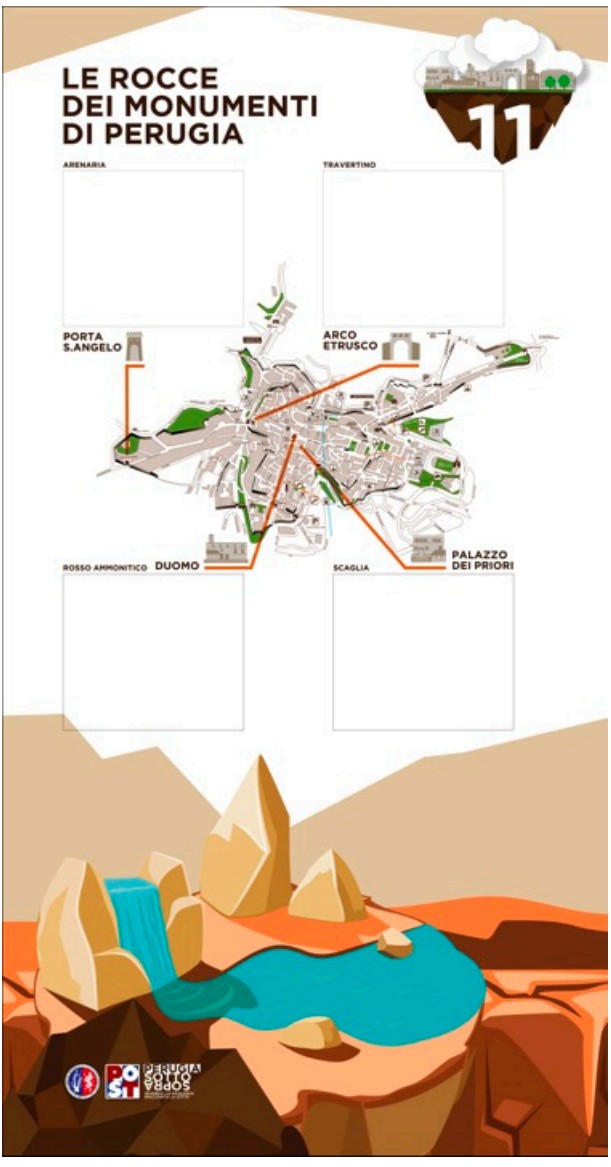

**Figure 4.** An example of a panel with the division of the space. In the upper part and at the bottom the graphics are present. In the middle part, the text and figures. In this example, the white squared contain the showcases with the rock samples referred to some historical buildings in Perugia. From the left corner on the top and proceeding clockwise: sandstone, travertine, limestone, Rosso Ammonitico Formation. The title of this panel is "Rocks and Monuments in Perugia" (the reproduction of the image is allowed by POST).

Multimedia tools interrupted the path of the exhibition, guided by the numbered panels. Transparent and illuminated showcases contained samples of rocks and terrain. Videos with real images and paleo-environments reconstructed with digital techniques were broadcast continuously. Moreover, some interactive tools invited visitors to create their own experience with the different geological components. In the opening period of the exhibition, some laboratories in the museum and outside were organized devoted to scholarships. Urban trekking completed the offer with the possibility for people of all ages and cultural levels to observe the places that they were introduced to in the exhibition within the city. The results and methods of this approach are described below.

## 2.2. Results

### 2.2.1. The Panels

The exhibition was structured in five sections all included in the main hall of the museum (Figure 5), each one devoted to a particular aspect of urban geology present in Perugia with a theoretical scheme following an initial introduction to the geological history and then moving toward some more particular aspects.

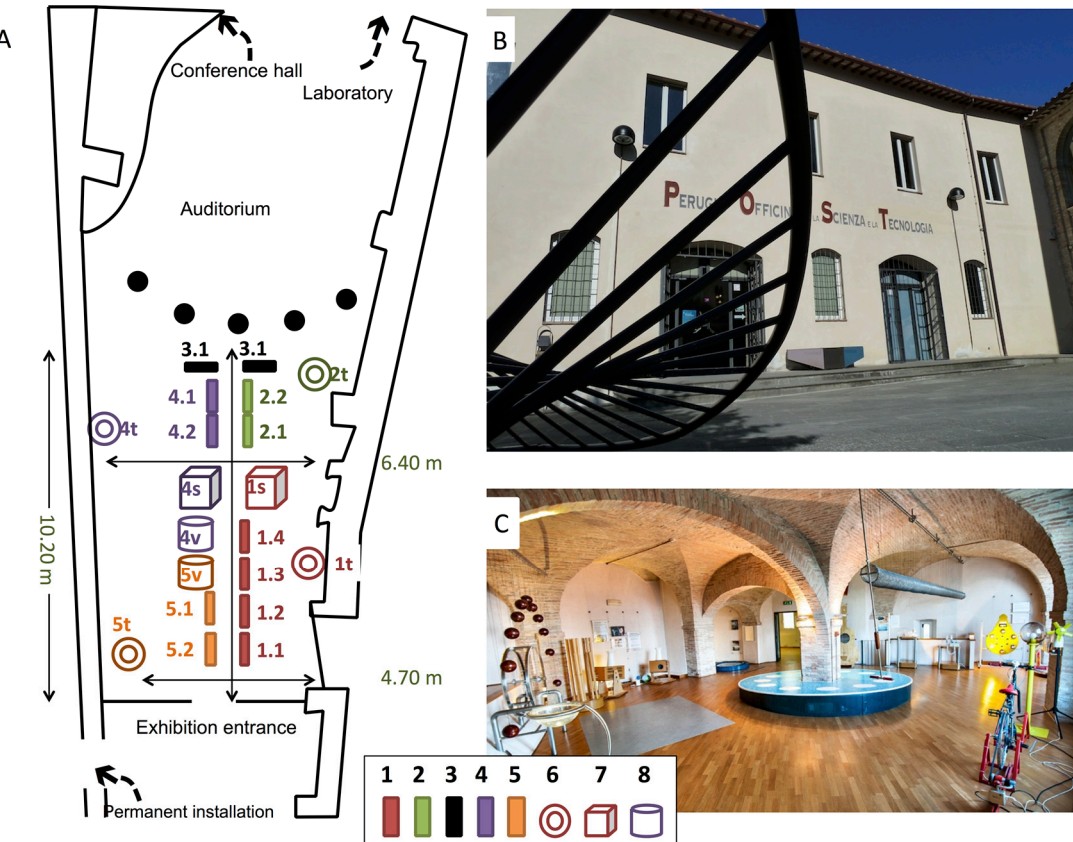

**Figure 5.** (**A**) Map of the POST museum. In addition to the room where the exhibition was installed, the museum has a room with a permanent installation, an auditorium, a conference hall and a room for the laboratories. (1) Panels of geological section, (2) panels of geomorphological section, (3) panels of the human presence, (4) panels of building stones section, (5) panels of paleontological section, (6) tools: (1t) 3D puzzle, (2t) ARSandbox, (4t) optical microscope, (5t) rhino skull model. (7) Showcases: (1s) boxes with conglomerates, sand, clay, (4s) boxes with samples of travertine, limestone, and sandstone. (8) Videos: (4v) video of building stones, (5v) video of Pleistocene paleoenvironments. (**B**) the entrance of POST museum, (**C**) The room with the permanent installation (photo by POST, use allowed by POST).

The first section was assigned to the geology, followed by the second one, where the geomorphology was the topic. The third section illustrated the relationship between human presence and geological context, while the fourth section was dedicated to building stones. The fifth section was used to house the paleontological heritage found in the city and to explain the fauna present in the Pleistocene (Figure 6).

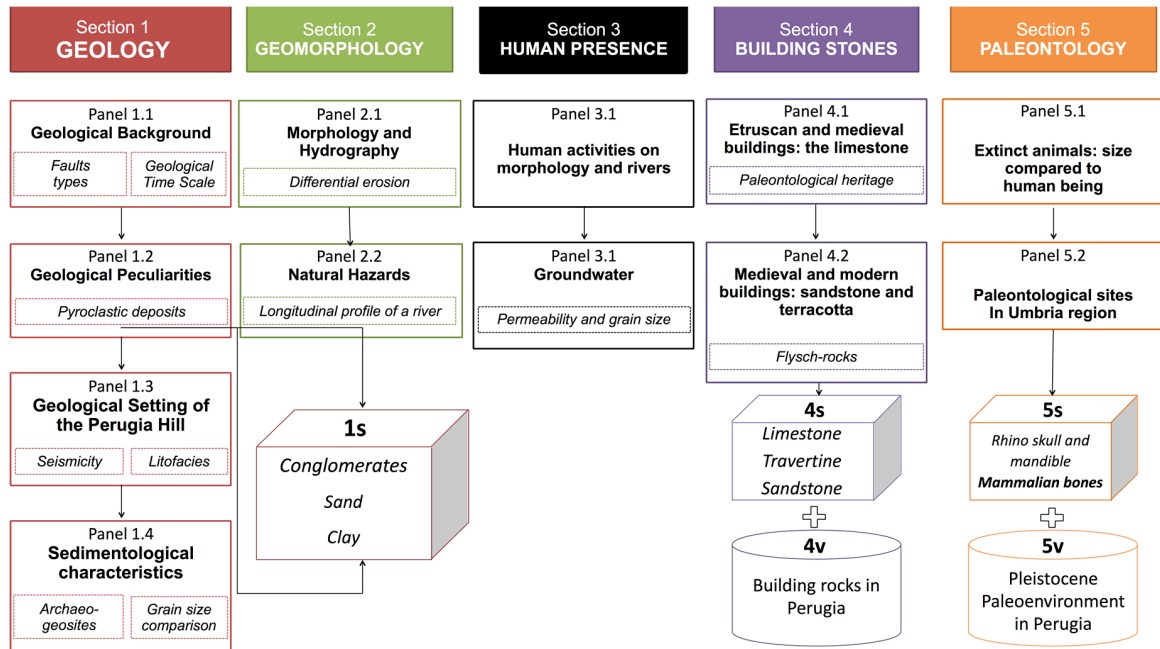

**Figure 6.** The list of the panels divided according to the different sections: geology, geomorphology, human presence, building stones, paleontology. The topic of each panel is under the number. The subjects of the in-depth boxes are specified in the frames with the dashed lines. The cubes represent the showcases (s) while the cylinders the videos (v).

The geological section (Section 1) proposed four panels where the bedrock composition and the geological evolution of the area were summarized (Figure 7).

In the panel 1.1 the geological background summarized, with a geological time scale, the entire geological history of Umbria region from the oldest rocks dated about 250 My up to now. Two in-depth boxes better explained what a fault is and the principles of stratigraphy. The panel 1.2 had the aim to dispel some "false myths" still deep-rooted in the popular culture of the place. In particular local traditions identified some mountains close to Perugia as ancient volcanoes. This information is still present in some websites, pointing out the poor communication between academic institutions and local people. The in-depth box tried to explain what is a true pyroclastic deposit. The panel 1.3 was focused on the geological setting of the Perugia hill with two in-depth boxes. The first one explained the concept of litofacies due to the fact that the sedimentary sequence outcropping in the city is organized in several litofacies. The second box illustrated the relationship of the area with the seismicity of the central Apennines. Although Perugia is located in an area with low seismic risk, moving eastward, the Apennines record events with high magnitude and thus the effects of seismic shocks are evident in the city too and affect, mostly from a psychological point of view, a large part of the citizenry. The first section ended with the panel 1.4 where the sedimentological characteristics of the deposits are detailed. Three showcases contained conglomerates, sand clay with a reference scale beside each box. Visitors were able to observe the difference in size between the various deposits. On the panel, one in-depth box suggested some archaeological sites in Perugia were these different deposits might be observed and introduce the concept of archaeo-geosite.

The second geomorphological section was split in only two panels. The first one (2.1) explains the relationship between morphology, hydrography, and the geological arrangement. The typical

landscape of Perugia, divided into ridges and rivers, has been interpreted with a geological approach. Due to the fact that some landscape particularities in the Perugia slopes are due to the different grain size of the deposits, the concept of differential erosion is detailed in a box. In the second panel (2.2) the attention was focused on the mass wasting and fluvial processes acting on the area with an analysis of related natural hazards. River erosion is the main cause of landslides, mostly along the headwater drainage divide close to the downtown. Therefore, the in-depth box explains the concept of the longitudinal profile of a river and the tendency to an equilibrium state, gained through erosion and sedimentation activities.

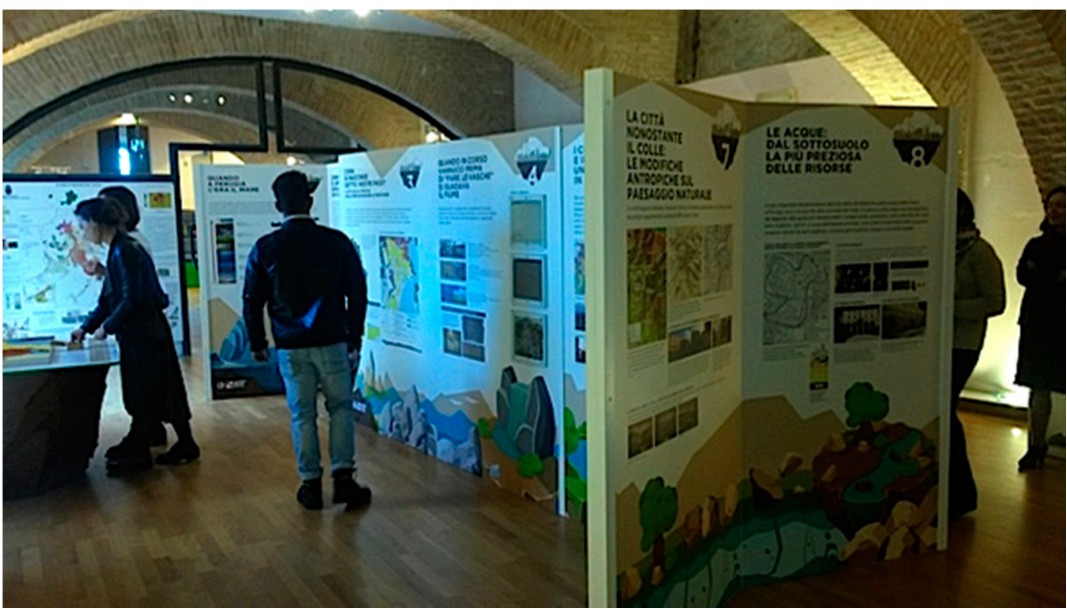

**Figure 7.** The exhibition along Section 1 geology and Section 2 geomorphology (on the left) and Section 3 human presence (on the right, photo by POST, use allowed by POST).

The third section is on the human presence and the two panels reveal the topographic surface changes made by humans over the centuries to prevent landslides or for the construction of important historic buildings. The definition of a "morphological false" is present in panel 3.1, to explain some characteristic areas in the downtown, and was very appreciated by visitors. The panel 3.2 highlights the ancient water supply methods. In the downtown a large number of historical wells and tanks, from the Etruscan (from V to I century B.C.) and medieval periods are present. Due to the sedimentary grain size sequence, the oldest part of the city has a huge amount of underground water reserve even today. In the panel the concept of porosity and permeability is detailed.

The mineralogical section is the fourth one and it was dedicated to the building stones. In fact, in Perugia, there is a very close relationship between some historical periods (Etruscan and Roman, medieval and the passage between the XIX and XX centuries) and the use of specific lithotypes for the construction of the main religious and civil buildings. The panel 4.1 shows the use of travertine in the Etruscan period (Etruscan walls) and of limestone in the medieval one, while the panel 4.2 highlights the use of sandstone in the medieval walls and of terracotta, derived from the clay present at the bottom of the hill, on the most recent historical buildings (beginning of XX century). The in-depth box reveals the paleontological heritage hidden on the façade of some important buildings in the downtown and that several tiles are made of Rosso Ammonitico Formation (Toarciano). The name of this formation, well widespread on the regional territory, derives from the high content of ammonite fossils. Four showcases contained many samples of travertine, limestone, and sandstone. A video with subtitles, close to the showcases, evidenced the use of these lithotypes on the most famous religious and civil buildings in the downtown of Perugia and the natural environments where these sedimentary rocks originate.

Finally, the last palaeontological section illustrates the mammal fauna of central Italy in the Pleistocene (Figure 8). One of the most important results of the exhibition was to show for the first time the mammal fossils (Pliocene and Pleistocene) discovered in the past century on the Perugia hill, with a well-preserved rhino skull usually not visible to the public. In this section, a video was present too (Figure 9). With surface mesh digital techniques some contemporary places in the city were overlaid with the moving images of mammal fossils in order to show the palaeoenvironmental conditions in the Pleistocene.

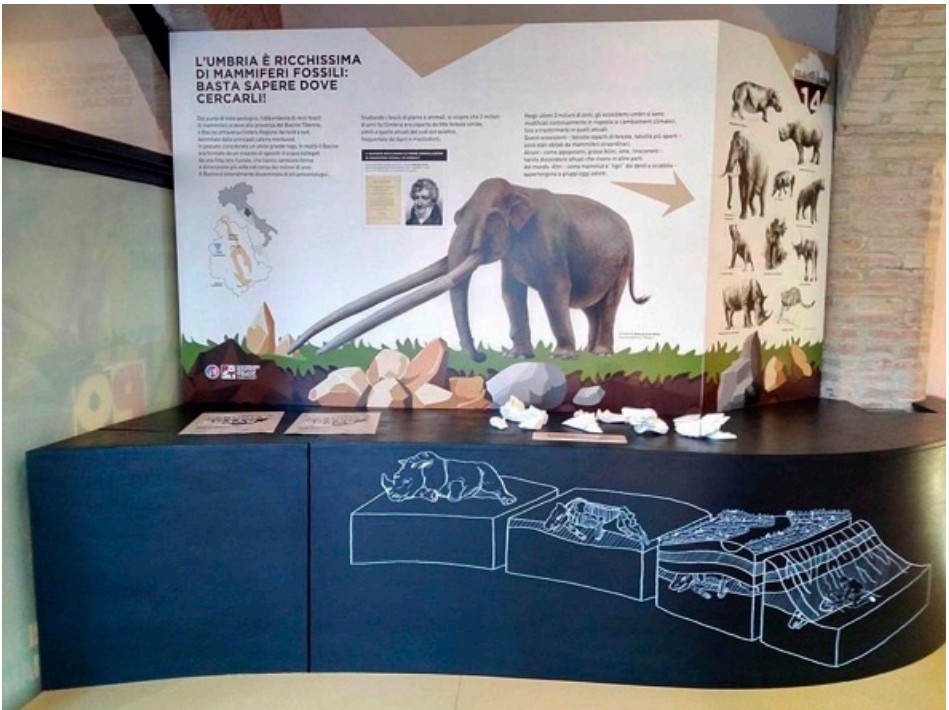

**Figure 8.** One of the panels in the paleontological section. On the desktop the several parts of the model of the rhino skull are visible (photo by L. Melelli, use allowed by POST).

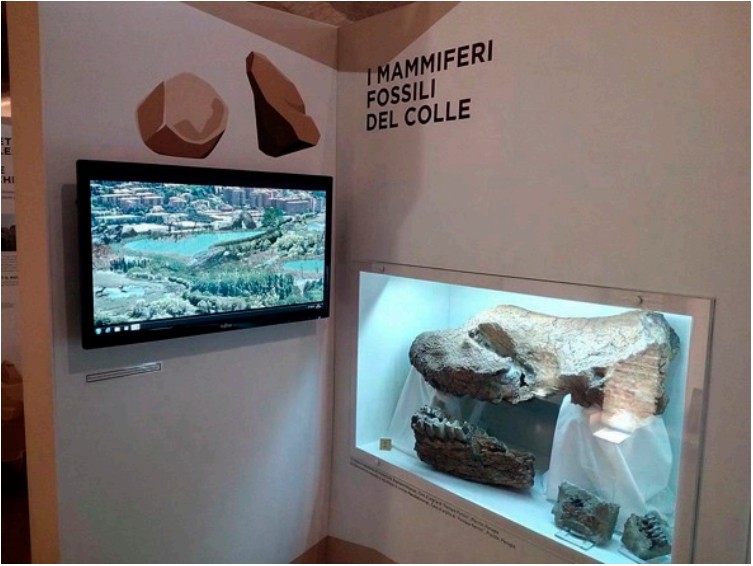

**Figure 9.** The paleontological section: the real rhino skull is in the showcase on the right, on the left the video 5v with a frame representing the merge between a present landscape of Perugia and a digital reconstruction of the lake present in the area in the Pleistocene with some mammals moving along the shore (photo by L. Melelli, use allowed by POST).

### 2.2.2. The Tools

In each section, a tool invited the visitors to be an active subject of the exhibition (Figures 5 and 10).

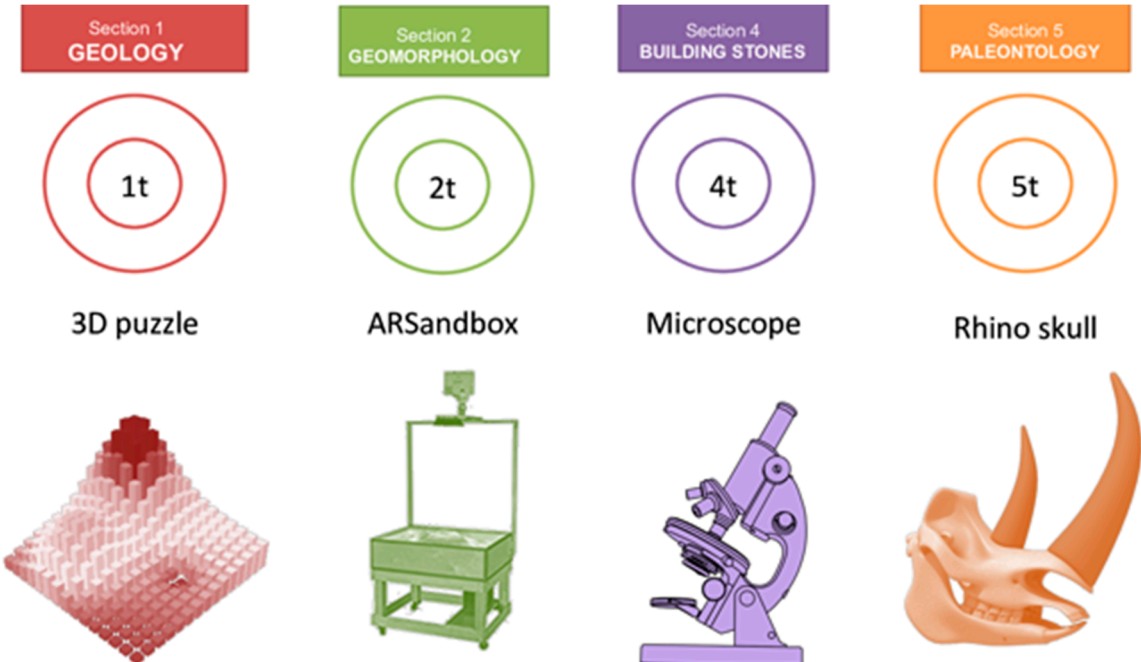

**Figure 10.** The tools in each section. The symbols with the numbers refer to Figure 5. 1t is the 3D puzzle in the geological section, 2t is the ARSandbox in the geomorphological section, 4t is the optical microscope in the section dedicated to building stones, 5t is the model of rhino skull in the paleontological section.

In the geological section, to help the visitor understand the spatial distribution of the lithotypes a 3D puzzle of the area was created (Figure 11). The first step was to extract some contour lines from a digital elevation model of the hill of Perugia (cell size 5 × 5 m). Then the polygons of the geological complexes were overlaid. Finally, only for the downtown area, the polygons of the watersheds are added where the drainage divide of the main rivers flowing on the city center converges. A 3D printer, analyzing the vector data, created the plastic model of the Perugia hill and surrounding area. Different colors were associated with the geological complexes while the plastic was cut along some boundaries corresponding to the limits between different lithological complexes or along drainage divides. Then some labels were available to be added to the puzzle and to identify the symbolic places of the city. In order to help the visitors, a poster in front of the plastic model was present with the names of the places printed on the labels.

In the geomorphological section, an augmented reality (AR) sandbox was installed (https://arsandbox.ucdavis.edu) allowing the 3D visualization of virtual topographic surfaces (Figure 12).

In particular, topographic contour lines and an elevation color map were visualized, and the water flow was simulated. The visitor, by hand-shaping the sand in the box, could modify the topographic surface and try to reproduce the morphology of the area.

In the mineralogical section, an optical microscope and thin sections of the main rocks present in the exhibition were made available to visitors (Figure 13). Each thin section was illustrated by a card where the petrographic and paleontological characteristics present in the thin section were detailed and highlighted. Beside the microscope, a hand lens was available for observing the macroscopic petrographic characteristics.

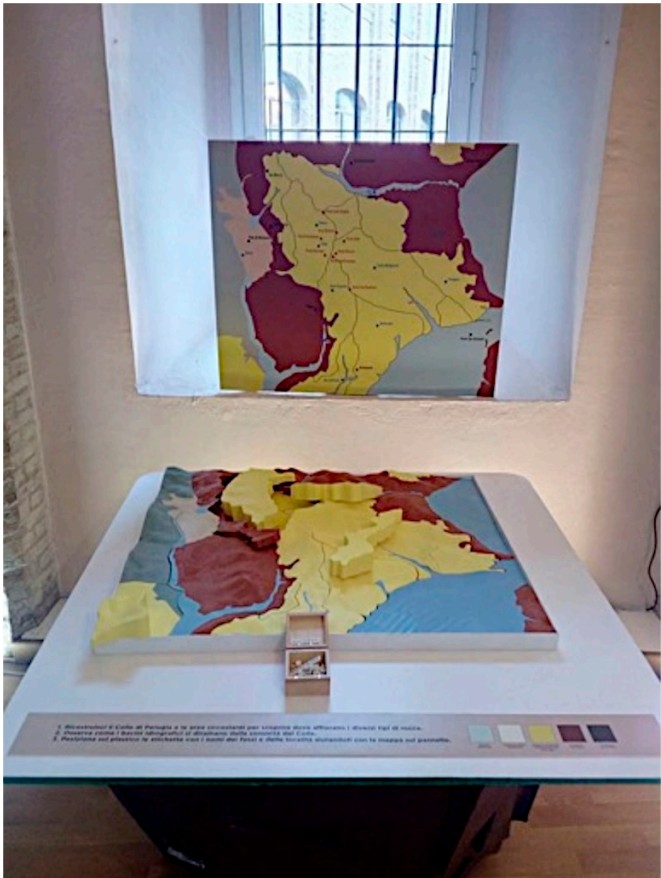

**Figure 11.** The 3D puzzle in the geological section. On the desktop, the model created with the 3D printer is available to visitors. The little box on the desktop contains the labels with the place names to be arranged on the model while the legend details the meaning of the different colors corresponding to the lithotypes. The poster hung in front of the window has the aim to help the visitors in doing this activity and represents the model in plain view with the watershed boundaries and the place names already put in order (photo by L. Melelli, use allowed by POST).

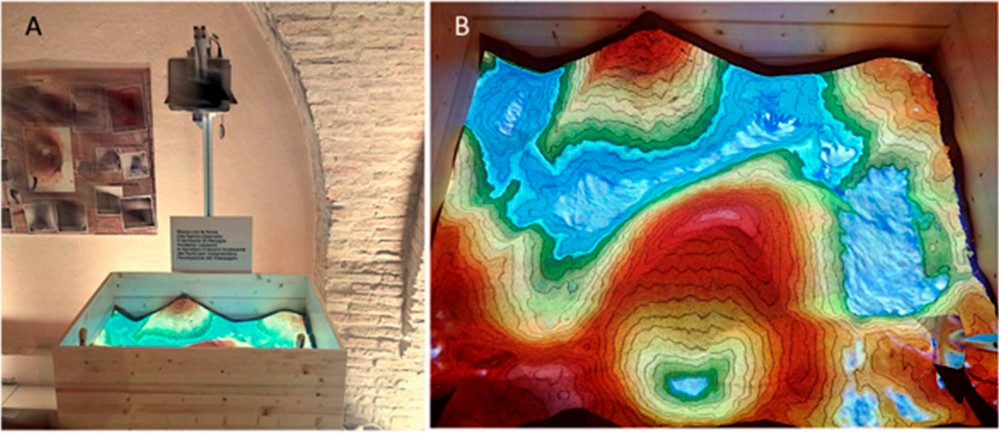

**Figure 12.** The ARSandbox in the geomorphological section. (**A**) The sandbox with the full equipment. (**B**) The surface of the model with the color ramp projected on the sand. The cold colors (blue one) refer to the lowest altitude, the heights increase going from green to yellow and brown for the highest altitude values. The contour lines are projected too. It is possible to observe in the hollowed areas the water effect (photo by L. Melelli, use allowed by POST).

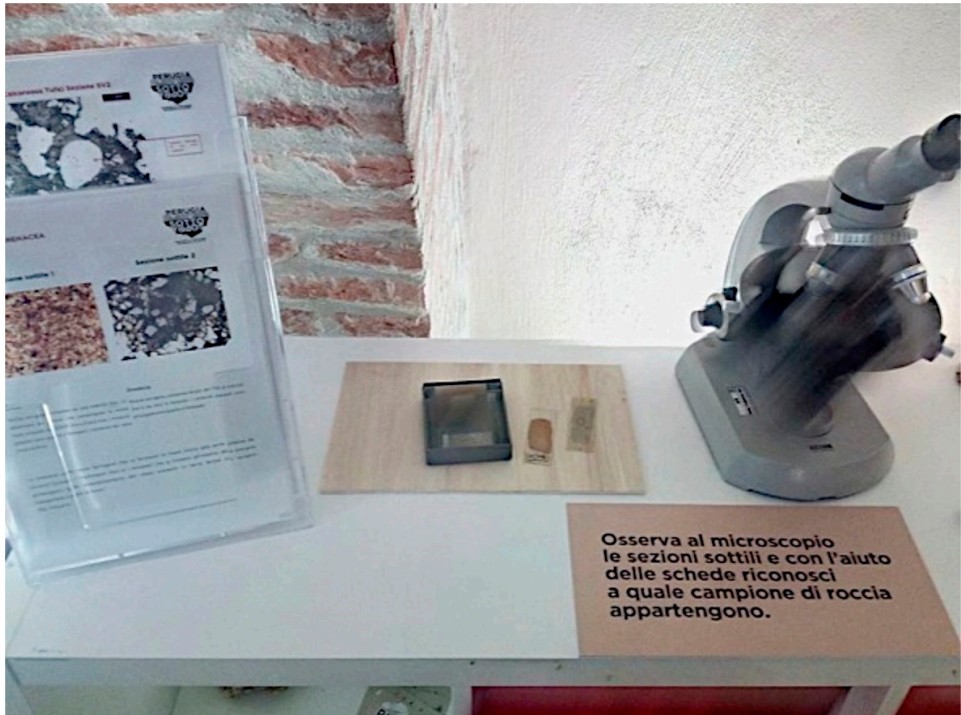

**Figure 13.** The optical microscope in the mineralogical section. On the left, the thin sections are available together with the instruction manual (photo by L. Melelli, use allowed by POST).

Finally, in the palaeontological section, a rhino skull was reproduced with a 3D printer and divided into some pieces along the morphological limits. Visitors were invited to put together the pieces to reconstruct the entire skull and better understand the shape and the function of the different pieces.

2.2.3. Laboratories and Trekking

During the regular time schedule for museum visits, some laboratories were organized. The laboratories were mainly dedicated to schools (Figure 14).

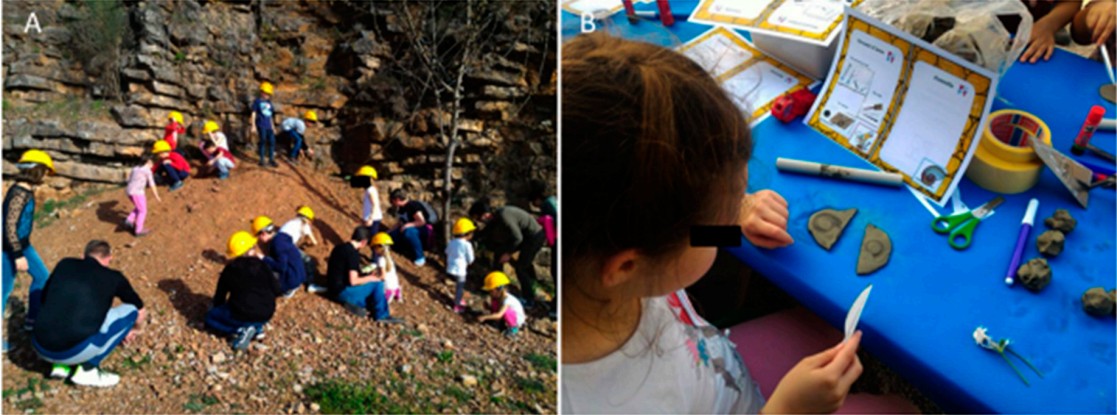

**Figure 14.** One of the laboratories prepared for the exhibition. In particular, this laboratory was dedicated to the paleontological section. (**A**) A school group working on the field to observe the rocks on an outcrop of limestone. (**B**) The laboratory's activity for creating the shape of some ammonites with modeling paste and for observing the morphological characteristics (photo by G. Margaritelli, use allowed by the author and by POST).

According to normal school planning, Earth sciences are focused on natural locations. In these laboratories, the aim was to introduce the cities as geological environments. Children, teenagers, and young people live daily in their cities, and most of their educational and recreational experiences are connected to urban infrastructures and places. For this reason, it is fundamental to exploit what each city can offer to bring young people closer to Earth sciences. Among the activities offered, the AR Sandbox appeared to be the most attractive tool. The key to understanding the scientific content is the augmented reality component. Contour lines and a terrain color ramp were projected on the virtual topography and movement was tracked using a Microsoft Kinect 3D camera. Placing an object at a particular height above the sand surface, a virtual rain is simulated, and water flowed over the landscape. Some fundamental topographic attributes, such as slope angle, could be visualized and easily modeled and modified. By connecting the slopes to the flow direction and accumulation may facilitate the understanding of drainage network modeling. Moreover, the AR Sandbox allows the capturing of photographs of the surface morphology at different times during use, rebuilding the sequence of events that modify the virtual landscape and offering the opportunity to follow its evolution over time. The strong point of this tool is that visitors can interact with the virtual topography by providing the SECRET "SEe and CREaTe" [46] for effective scientific communication. During the exhibition, weekly workshops were organized for schools of all levels and adult people (Figure 15A). Moreover, the material presented in the exhibition represented an important resource to be used in the activities of dissemination and information about the degrees in geology offered by the University of Perugia to different schools in the city.

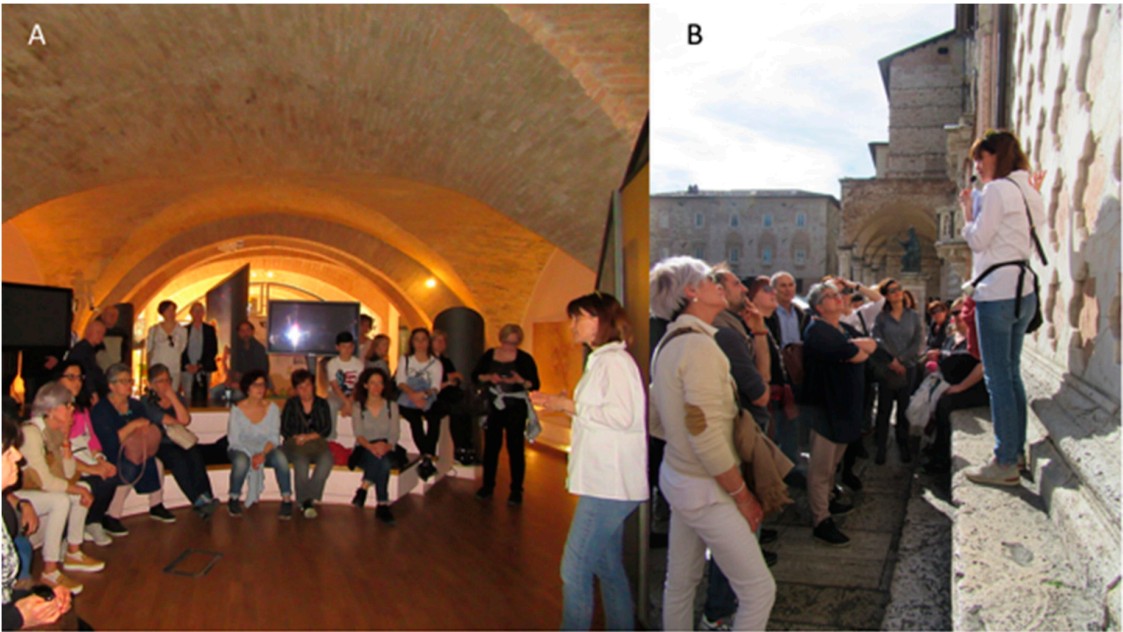

**Figure 15.** Some images of the activities organized during the exhibition. (**A**) A conference in the auditorium (see Figure 5), (**B**) trekking in the downtown to show the paleontological heritage on the building stones (photo by M. Coli and http://www.circolosanmartino.unipg.it, use allowed by the author and by POST).

Moving outside of the exhibition and remembering the information acquired inside the museum allowed visitors to complete their experience and to consolidate their cultural experience. The idea was to propose trekking tours in four dimensions (Figure 15B).

Two dimensions were presented walking along a path and referring to a map for improving the sense of direction and spatial arrangement of places. The third dimension was the perspective observable from scenic viewpoints. Being a hilly city, Perugia offers several scenographic standpoints. Moreover, Perugia has two opposite landscapes, the steep and uninhabited scenery along the northern

area and the gentle and urban one on the opposite side. This contrast is a good starting point for recalling geological and geomorphological aspects, such as tectonics, differential erosion, and fluvial and gravitational processes.

One of the most successful trekking routes was from the POST Museum up to the top of the downtown. There were six stops in total: one focused on the fluvial processes and natural phenomena, two on the anthropic modifications of natural morphology, two on building stones, and the last was run underground and exploited one of the most important Etruscan wells, the most important archaeological evidence of the ancient human presence on the hill related to water resources research. Trekking experiences represent the key to effective scientific communication. People could see, touch, look for, and most of all, connect an abstract idea to something tangible. Moreover, they could apply a scientific subject to daily life and acquire the capability to observe the urban environment from a different perspective. During the trekking tours, visitors were entertained above all by "fossil hunting". None of them, despite having lived in Perugia for decades, had ever noticed that on the facade of the city's main church, fossils of ammonites were present (Figure 16). This hints that the idea of the geologist obliged to search for scarce and rare fossils in natural environments is outdated, suggesting it is sufficient simply to observe our surroundings, especially those of historical buildings.

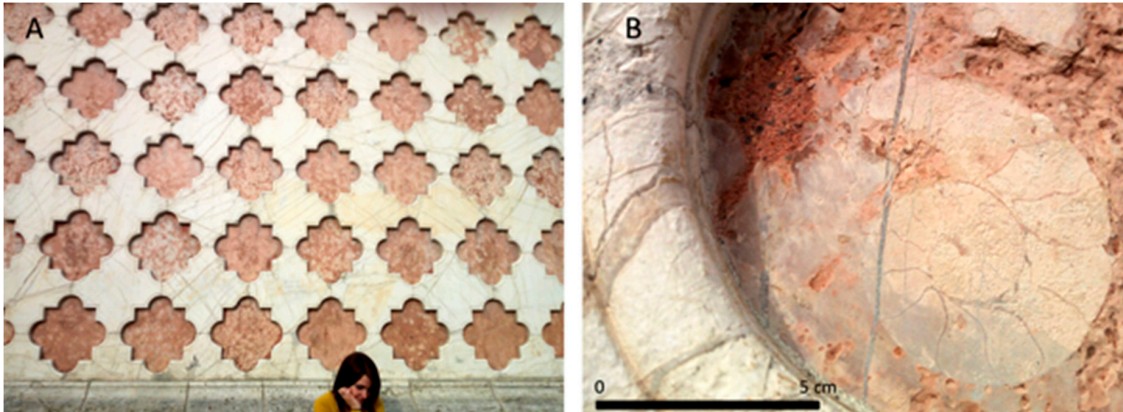

**Figure 16.** The palaeontological heritage on the building stones: (**A**) The façade of the San Lorenzo Cathedral, the most important church in the Perugia city, also in Figure 15. The limestone shows two colors, pink and white, for a better aesthetic result. (**B**) One ammonite inside the tiles.

*2.3. Discussion*

Urban geotourism is a promising approach to disseminating Earth sciences to a wide audience. Urban areas guarantee several advantages compared to natural environments. Cities with relevant historical and artistic contexts are generally already structured for needs related to tourism. The connection between human activities and the original natural environment, both in past centuries and in the present day, is well evident. Cities are places where digital tools (Wi-Fi and electronic devices, such as smartphones and tablets) are, in most cases, already structured and available for free [47] so that in urban areas the dissemination activities are facilitated and encouraged in order to increase the tourist flow. Several approaches are already tested in several cities in the world [48]. São Paulo in Brasil [49], Mexico City [30], London (http://londonpavementgeology.co.uk), Lisbona in Portugal [31], Brno city in Czech Republic [50], Belgrad in Serbia [34], Shiraz city in Iran [32] are only some examples. In Italy the geotouristic approach in urban areas has been already tested in some important cities. Rome [41,42,51,52], Milan [29], Genoa [53], Naples [54], Turin [55].

The "Perugia Upside-Down" exhibition was the first experience of geotourism dedicated to urban geology in the city of Perugia. 8046 people, 3915 of whom were students, visited the exhibition. This number is a good result for the city and an excellent outcome for the POST Museum, which is dedicated exclusively to scientific topics. Panels, real samples in showcases, videos,

and multimedia tools are the avenues chosen to involve the public in themes present in the exhibition. Didactic laboratories and urban trekking are an incisive answer to "force" the visitors in moving out from the museum and discovering the contents of the exhibition in the real world. Moreover, Perugia, if compared with other cities like Roma or Milan, has the great opportunity to be in a hilly environment. Trekking activities may exploit several scenic views and the geomorphological experience could be much more interesting and richer.

The exhibition, despite good results, made clear some critical issues. The structure and content of the panels fully satisfied visitors. However, the number of panels and the large amount of information within them has made it difficult for younger visitors to understand. The texts should be written with non-technical terms, but in particular, they should be extremely brief. Although the tools obtained the best results in terms of involvement, two of them have raised some problems. In particular, the optical microscope showed significant limits. The managing of the several mechanical and optical components of the microscope requires a specialist beside the visitors. Although an explanatory sheet was next to the microscope, the comprehension of the thin section was not always clear. The location of the microscope was a mistake too, being along the path and without a dedicated corner where the visitors could observe the thin section comfortably and without feeling rushed. An alternative method, like a screen connected to the microscope with predefined focus, guided views and only some controlled rotation of the objects could be an alternative and better solution. The 3D model of the rhino skull was not always easy to manage for the visitors. The model was divided into some parts, according to a morphological principle. When the visitors found the sections already divided on the desk, it was very difficult to put the model together again. A detailed guide with the instructions listed step by step and figures of each component could facilitate a better understanding of the procedure. The most successful tools were the 3D puzzle and the ARSandbox. In both cases, no difficulty was identified. The visitors presented themselves as both amused and interested. These results confirm that when the dissemination activity satisfies the SECRET (SEe and CREaTe) for good communication, it goes beyond the limits imposed by the scientific nature of the content. The 3D puzzle is particularly worthwhile for obtaining awareness of geographical space and acquiring the ability to orientate places and put them in topological relation. The third dimension of the model facilitates the understanding of the distribution of altitude values. Observing and touching the distribution of slope values makes it possible to link some theoretical concepts, such as river erosion and the connection with slope evolution. In addition, the lithotypes being highlighted with different colors, it is possible to explain the influence of structural factors on superficial morphology. The ARSandbox is efficient in communicating the concepts of geomorphological processes, in particular, where the runoff is the main focus. The contour lines being visualized together with a color ramp make the sandbox a perfect visualization tool in the modeling of the real world with topographic maps. Moving the sand, the visitors modify the topographic surface and control the topographic attributes like slope, aspect, and curvature. The superimposition of the water flow effect shows the interaction between river drainage network and topography. The advantage of ARSandbox is the strong interaction opportunity presented to the visitors with the tool, mostly effective with young people and children.

Finally, for urban geoheritage promotion, the trekking experience turned out to be extremely positive. Visitors were invited to express their opinion and the results were extremely positive. Once again, to combine the daily experiences in the real world with theoretical concepts seems to be the key for effective dissemination of urban geological phenomena. Despite this, if compared with other similar experiences, the trekking activity could be improved. If urban areas offer some advantages in using digital technologies, this possibility should be strongly exploited. Where digital technologies empower the tools for geotourism, new approaches and potentialities are growing. This is the case of the mobile application technology developed for Lausanne [55], Turin [54], and Rome [55]. In the "Perugia Upside-Down" trekking activities the structure of the trekking was the traditional one with a guide speaking in front of the point of interest. This simple solution is not the most charming and the introduction of a mobile application is strongly recommended.

### 3. Conclusions

In 2017, looking for the best practice to transfer knowledge from a scientific or technical community to a broader audience in an urban environment, the Department of Physics and Geology in the University of Perugia organized an exhibition. The idea was to open decades of data collected by geologists, archaeologists, historians, and architects to citizens and tourists. The exhibition was structured in panels, interactive tools, laboratories, and trekking within the city. In this video: https://www.youtube.com/watch?v=oDng-kPKvpw, it is possible to take a virtual tour of the exhibition. The experience, despite good results, highlighted some critical issues. In the panels, the text could be further shortened and simplified. Some tools turned out not to be suitable for an exhibition for educational purposes or, more precisely, not without some precautions that simplify their use. Trekking in urban areas could be more effective if supported by digital devices that expand the information.

Starting from the "Perugia Upside-Down" experience, new projects started in order to improve geotourism. SILENE (a LIDAR system for exploring the Palazzone necropolis remote sensing and geology for enhancing archaeological sites) is a project with the aim of promoting the Etruscan necropolis of Palazzone in Perugia, that is undoubtedly one of the most valuable Etruscan burial sites in Central Italy [26,45]. More than two hundred tombs are present in the necropolis, all dug at different levels within the deposits of the Perugia hill. The perimeter walls are real "three-dimensional geological sections", allowing the observation of the sediments from various orientations. The project revealed to the visitors the paleogeographic environment of the Perugia hill through the sedimentary structures present in the deposits suggesting the importance of the Necropolis as an archeo-geosite where historical-artistic value and geological importance are combined. GPS and digital surveying (LIDAR—laser imaging detection and ranging) together with a drone appropriately equipped for carrying out aerial surveys, allowed topographic maps, orthophotos and a detailed digital model assisting in the production of virtual images and tours. The results are visible on http://www.silenepg.it.

The experience of urban trekking during the exhibition suggests us to exploit digital techniques to better involve people in consuming information and obtaining a completely satisfying experience. For this reason, a second project is being developed, named HUSH (hiking in urban scientific heritage). Mixing science, technology, and augmented reality, HUSH will show the naturalistic and geological heritage hidden in the city along several urban trekking routes. The recent advancements in augmented reality technologies create the basis for the development of immersive and customized touristic experiences (abstract HUSH). The last ongoing project is HUSH Underground that is a section of HUSH dedicated to the underground cavities present in the downtown area of Perugia. The common starting point of all these projects is the geological heritage hidden in the city of Perugia. To this day, geology in Perugia has been linked to the hydrogeological instability affecting large areas close to the downtown. With this new approach, geotourism could be a precious resource and a unique opportunity not only for future research but for didactic and cultural purposes with significant commercial and administrative impacts.

**Funding:** This research was funded by the Dipartimento di Fisica e Geologia, Università degli Studi di Perugia, project title "PERUSIAE (PERUgia StratIgraphy, geoArcheology and landscapE): a multidisciplinary reappraisal of the geological assessment of Perugia Hill)", RicBAs2014, awarded to Laura Melelli.

**Acknowledgments:** The author would like to thank POST Museum (http://www.perugiapost.it) for the contribution in the creation of the structure and organization of the exhibition. The staff of the communication agency "Le Fucine Art & Media" (https://www.lefucine.it) are authors for the design of the panels and for the poster used for the advertising. Marco Cherin (https://www.unipg.it/personale/marco.cherin) is the author of the paleontological section. The Sabrina Nazzareni (https://www.unipg.it/personale/sabrina.nazzareni) is the author of the description of the thin sections used with the optical microscope.

**Conflicts of Interest:** The author declares no conflict of interest.

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
