# Peer review of "“Perugia Upside-Down”: A Multimedia Exhibition in Umbria (Central Italy) for Improving Geoheritage and Geotourism in Urban Areas"

_resources, doi:10.3390/resources8030148_

Round 1
Reviewer 1 Report
General comment:
I think that this type of paper, where the authors describe an exhibition and its activity needs more images of all the contents and experiences. Also some tables where the author resume the main lines of the exhibition are necessary.
Tittle: include the site where the paper is focused
Line 82. A paragraph with only 2 lines?, please restructure this the different parts of the paper. Try to construct paragraphs/ideas with several lines, not only one line.
Figure 1. Geological map in the context of Italy, or Mediterranean (please add a more bigger scale map to locate the study area). Why not include a geological map of the study area?, equivalent to this geographical map.
Line 190. Only one line?, please restructure
I suggest to include a table, with the main topics/ideas of the sections, panels and tools to follow the text
Figure 2. It is not possible to read the text
I suggest to incorporate more photos (detailed images) of the exhibition, and why not construct figures with some images (a,b, c…)
3.2. ; I suggest to the describe with more detail this point. Also include a table with the different ideas of the workshop. Is there any photos of the workshops developed?
Figure 7. Scale in the right image?
Author Response
Response to Reviewer 1 Comments
I would like to thanks to the Reviewer for the advices. I completely agree that the paper should be improved.
I strongly improved the text, the figures, the captions and the references.
A native speaker checked the paper.
Point 1: General comment:
I think that this type of paper, where the authors describe an exhibition and its activity needs more images of all the contents and experiences. Also some tables where the author resume the main lines of the exhibition are necessary.
Response 1: I strongly modified the article: the text, the figures and the captions. They are now different in number and content. The reference list has been considerably expanded.
Point 2: Tittle: include the site where the paper is focused
Response 2: The title has been updated and now includes the location of the paper.
Point 3: Line 82. A paragraph with only 2 lines?, please restructure this the different parts of the paper. Try to construct paragraphs/ideas with several lines, not only one line.
Response 3: Also the reviewer 2 suggests a meaningful modification of the text. So that now it is quite different and also this suggestion was done.
Point 4: Figure 1. Geological map in the context of Italy, or Mediterranean (please add a more bigger scale map to locate the study area). Why not include a geological map of the study area?, equivalent to this geographical map.
Response 4: Now there are two figures in the beginning of the paper and more information have been added.
Point 5: Line 190. Only one line?, please restructure
Response 5: It was done.
Point 6: I suggest to include a table, with the main topics/ideas of the sections, panels and tools to follow the text
Response 6: In order to better follow the text and the organization of the exhibition figg. 5, 6 and 10 have been included.
Point 7: Figure 2. It is not possible to read the text
Response 7: the figure was completely reviewed.
Point 8: I suggest to incorporate more photos (detailed images) of the exhibition, and why not construct figures with some images (a,b, c…)
Response 8: It was done.
Point 9: 3.2. ; I suggest to the describe with more detail this point. Also include a table with the different ideas of the workshop. Is there any photos of the workshops developed?
Response 9: It was done..
Point 10: 3 Figure 7. Scale in the right image?
Response 10: It was added.

Reviewer 2 Report
This article is based on an in-depth research, and it will be highly-interesting to the wide circle of specialists. It can be accepted after certain improvements. My recommendations are listed below.
1) Please, do not capitalize words Geology, Geoheritage, etc. like in the first line of Abstract (please, check the rest of the text too).
2) Speaking about components of geotourist resources, why not to consider 1) building stones as geoheritage and 2) tall buildings/bridges/industrial constructions as viewpoint geosites (sensu Migon and Pijet-Migon, 2017 in Proceedings of the Geologists' Association and Mikhailenko and Ruban, 2019 in Land – cite these works, please).
3) Introduction does not state the objective of the present paper and does not inform about its geographical focus.
4) The number of the cited literature sources should be extended by 10-20 items. 1) Many conceptual papers on urban geoheritage and geotourism has been published in 2018-2019. Please, try to collect this information from Scopus or ScienceDirect (e.g., check new papers by E. Reynard, T. Habibi, etc.). These papers should be cited in Introduction for general reference. 2) There are many books and articles reviewing the issue of urban geology, including the classical works of Leggett and Karrow. Why not to cite these?
5) Section 3 is informative. But it should be split into Methodology, Results, and Discussion. A lot of relevant additions should be made. Discussion should also bear comparison of the urban geotourism practice in Perugia and other cities, including those outside of Italy (e.g., check the excellent works of Del Lama on Sao Paulo).
6) Section 4 should be named Conclusion. Much information from there should be moved to the previous section, whereas Conclusions should list the main outcomes of the present study and state perspectives for future investigations.
7) Why [32] occurs separately on the last line of the main text? The paper should not end with a citation.
8) In Acknowledgements, the author states this paper is a result of teamwork. Ok, but where are the other members of the team? Why these are not co-authors?
9) Figure 1: is this is an own drawing of the author? If not (I guess this is the official map published by any geological survey), please, give a citation.
10) I'd like to be sure that the author has a permission to publish photos from museums (this should be stated in Acknowledgements or any other technical section).
11) The captions of many figures are too brief. These does not inform about where the photos were captured. A good solution would be to add a simple map of Perugia and to indicate all localities mentioned in the text and shown on photos.
Good luck with revision!
Author Response
Response to Reviewer 2 Comments
I would like to thanks to the Reviewer for the advices. I completely agree that the paper should be improved.
I strongly improved the text, the figures, the captions and the references.
A native speaker checked the paper.
Point 1: 1) Please, do not capitalize words Geology, Geoheritage, etc. like in the first line of Abstract (please, check the rest of the text too).
Response 1: It was done
Point 2: 2) Speaking about components of geotourist resources, why not to consider 1) building stones as geoheritage and 2) tall buildings/bridges/industrial constructions as viewpoint geosites (sensu Migon and Pijet-Migon, 2017 in Proceedings of the Geologists' Association and Mikhailenko and Ruban, 2019 in Land – cite these works, please).
Response 2: In the first submitted version of the paper the idea of building stones as a geotouristic resource was suggested (see lines from 88 – 90 “Presently an opposite trend is growing in the scientific and administrative environments: the geological context as a new and promising resource for the touristic and didactic issues in urban areas. One of the most successful approaches is the heritage stones used for historical buildings”).
The value of building stones as geoheritage and the suggested works have been added in the revised text.
Point 3: 3) Introduction does not state the objective of the present paper and does not inform about its geographical focus.
Response 3: In the Author’s idea the introduction should introduce the aim of the paper in a broad perspective. Therefore the concepts of geotourism and geoheritage are illustrated, with particular attention to the problems and possibilities of the dissemination activities. However the Author improved the Introduction with other considerations about scientific exhibitions and with the geographical focus, as suggested by the reviewer.
Point 4: 4) The number of the cited literature sources should be extended by 10-20 items. 1) Many conceptual papers on urban geoheritage and geotourism has been published in 2018-2019. Please, try to collect this information from Scopus or ScienceDirect (e.g., check new papers by E. Reynard, T. Habibi, etc.). These papers should be cited in Introduction for general reference. 2) There are many books and articles reviewing the issue of urban geology, including the classical works of Leggett and Karrow. Why not to cite these?.
Response 4: In the first submission the references were 32. In this revised version the references are 57. All the suggested papers have been included.
Point 5: 5) Section 3 is informative. But it should be split into Methodology, Results, and Discussion. A lot of relevant additions should be made. Discussion should also bear comparison of the urban geotourism practice in Perugia and other cities, including those outside of Italy (e.g., check the excellent works of Del Lama on Sao Paulo).
Response 5: The section 3 has been splitted, as suggested, in methodology, results and discussion. The entire paragraph has been strongly improved.
Point 6: 6) Section 4 should be named Conclusion. Much information from there should be moved to the previous section, whereas Conclusions should list the main outcomes of the present study and state perspectives for future investigations.
Response 6: Section 4 is now named “Conclusion” and it was modified as suggested.
Point 7: 7) Why [32] occurs separately on the last line of the main text? The paper should not end with a citation.
Response 7: It has been modified.
Point 8: 8) In Acknowledgements, the author states this paper is a result of teamwork. Ok, but where are the other members of the team? Why these are not co-authors?
Response : The Acknowledgements has been rewritten. The reason of a single Author is detailed below.
The exhibition is the result of years of research carried out by the author without external collaboration.
The IDEA of the exhibition was exclusively by the author.
The PATH and the SEQUENCE of the panels are exclusively from the author.
The TEXTS in the panels are exclusively from the author. They were only revised and reduced by the team in the museum.
The ORIGINAL FIGURES are exclusively from the author, the PHOTOS and MAPS are the result of the bibliographic research of the author, collected in some years prior to the exhibition. In the panels all the credits were cited.
The TOOLS are an idea of the author.
The only two exceptions are the following:
The PALEONTOLOGICAL SECTION was entirely realized by Dr. Marco Cherin, as cited in the Acknowledgements. The description of the thin sections, synthesized as guides next to the optical microscope were written by Dr. Sabrina Nazzareni, as cited in the Acknowledgements.
The team mentioned in the acknowledgements is referred to the museum staff and graphic designers.
Nevertheless, since the scientific effort for the realization of the exhibition was almost exclusively by the Author, the Author is the unique writer of the article.
Point 9: 9) Figure 1: is this is an own drawing of the author? If not (I guess this is the official map published by any geological survey), please, give a citation.
Response 9: The Figure 1 (now figure 2) is completely drawn by the Author. It is created in a GIS environment. Moreover in the first version of the paper the figure is not a geological map but a Digital elevation Model (free data and in any case modified in GIS). The geological map added in this new version is an original work by the author based on the official geological database of the Umbria region. The polygons in the database, downloadable for free, were reworked by the Author and the final image is an original work.
Point 10: 10) I'd like to be sure that the author has a permission to publish photos from museums (this should be stated in Acknowledgements or any other technical section). Response 10: The Author has the written permission granted by the Museum Director to publish photos from the museum as declared in the Acknowledgements. If it is necessary I can provide the permission. In all the captions with photos of the museum or related to the activities during the exhibition the Author has added the permission of POST and the name of the author of the photos.
Point 11: 11) The captions of many figures are too brief. These does not inform about where the photos were captured. A good solution would be to add a simple map of Perugia and to indicate all localities mentioned in the text and shown on photos.
Response 11: The captions are rewritten for a better comprehension of the figure. Since there is only a figure referred to a place in Perugia (the ammonites on the façade of the San Lorenzo Cathedral) it seems not very useful to add an image with the single localities. Moreover in this revised version of the paper the figure 2 includes several place name around downtown.

Round 2
Reviewer 1 Report
-
Reviewer 2 Report
I'm very satisfied with the author's constructive reaction to criticism, in-depth work, and improvements. The manuscript looks nice now. I tend to recommend its acceptance.